# Wound Healing versus Metastasis: Role of Oxidative Stress

**DOI:** 10.3390/biomedicines10112784

**Published:** 2022-11-02

**Authors:** Tatiana Lopez, Maeva Wendremaire, Jimmy Lagarde, Oriane Duquet, Line Alibert, Brice Paquette, Carmen Garrido, Frédéric Lirussi

**Affiliations:** 1UMR 1231, Lipides Nutrition Cancer, INSERM, 21000 Dijon, France; 2UFR des Sciences de Santé, Université Bourgogne Franche-Comté, 25000 Besançon, France; 3Plateforme PACE, Laboratoire de Pharmacologie-Toxicologie, Centre Hospitalo-Universitaire Besançon, 25000 Besançon, France; 4Service de Chirurgie, Centre Hospitalo-Universitaire Besançon, 25000 Besançon, France; 5Centre Georges François Leclerc, 21000 Dijon, France

**Keywords:** wound healing, metastasis, oxidative stress, macrophage, Hypoxia Induced Factor, Nuclear Factor Kappa B, nuclear factor erythroid-2-related factor 2

## Abstract

Many signaling pathways, molecular and cellular actors which are critical for wound healing have been implicated in cancer metastasis. These two conditions are a complex succession of cellular biological events and accurate regulation of these events is essential. Apart from inflammation, macrophages-released ROS arise as major regulators of these processes. But, whatever the pathology concerned, oxidative stress is a complicated phenomenon to control and requires a finely tuned balance over the different stages and responding cells. This review provides an overview of the pivotal role of oxidative stress in both wound healing and metastasis, encompassing the contribution of macrophages. Indeed, macrophages are major ROS producers but also appear as their targets since ROS interfere with their differentiation and function. Elucidating ROS functions in wound healing and metastatic spread may allow the development of innovative therapeutic strategies involving redox modulators.

## 1. Introduction

The process of wound healing is a successive well-organized cascade of events involving specific cellular and molecular actors with the intent to restore tissue homeostasis and protect it from infection. On the contrary, unsuccessful healing is associated with severe clinical outcomes such as tumor development. It is also now well documented that some wounds like diabetic wounds or septic injury are associated with tumor progression and/or with an increased risk of cancer relapse.

Excessive and prolonged inflammation during inadequate wound healing process creates a microenvironment that shares strong similarities with tumor stroma. Notably, these microenvironments are characterized by hypoxia that generates a neovascularization for nourishment, influx of leukocytes sustaining the inflammatory response and breakdown/remodeling of the extracellular matrix. The tight similarity between wounds and tumor stroma generation have been first proposed by Rudolph Virchow in 1858 with his ‘irritation theory’, in which he concluded that irritation and its subsequent inflammation were the essential factors that led to the formation of neoplastic tissues [1]. Over a century later, these same similarities have led Harold Dvorak to state that tumors are ‘wounds that do not heal’ [2]. The strong similarities between the wound healing process and metastasis dissemination have been extensively and recently reviewed elsewhere in particular concerning the pivotal role of inflammation. Nevertheless, inflammation is also associated with oxidative stress either via Reactive Oxygen Species (ROS) and/or via Reactive Nitrogen Species (RNS) production that may play a key role in the clearing repairing process. Surprisingly, while these mechanisms are well known for wound healing, it remains poorly studied concerning metastasis. In this mini-review, we will focus on the specific contribution of oxidative stress in these two (physio)-pathological processes, in particular by describing the molecular and cellular actors involved. Studying pathophysiological mechanism of wound healing may help to better understand the metastasis process and lead to new therapies and vice versa.

## 2. Good and Bad Wound Healing: Acute versus Chronic and Chronology of the Cellular Actors

Four distinct and overlapping steps are needed in the physiological wound healing process: (1) hemostasis, (2) inflammation, (3) new tissue formation and (4) tissue remodeling [3,4,5].

Hemostasis consists of the formation of platelet plug, blood clot and consequent local hypoxia. Then, during the inflammation phase, neutrophils and tissue resident macrophages are the first immune responding cells to the wound [6,7,8,9]. This stage also induces immune cells invasion particularly monocytes recruited from the bone marrow and differentiated into mature inflammatory macrophages (named M1) [10]. Proteolytic enzymes, pro-inflammatory cytokines, growth factors and ROS are secreted [11] to protect organism against bacterial or other micro-organisms invasion. After this step, the levels of pro-inflammatory cytokines and oxidative stress decrease to return to a basal state [11]. Resolving anti-inflammatory macrophages (named M2) contribute to remove cells and bacteria debris by efferocytosis or phagocytosis [12]. Keratinocytes, fibroblasts and endothelial cells migrate to the wound and proliferate to initiate new tissue formation stage. Finally, tissue remodeling macrophages promote matrix metalloproteinase (MMP) expression in order to restore functional and anatomical integrity of tissue (Figure 1) [13,14,15,16].

After injury, various cells are recruited during the early phase of wound healing. During the inflammation stage, platelets first migrate to the site of the injury to induce coagulation followed by neutrophils. At the same time, ROS level increases while local oxygen concentration decreases leading to hypoxia. Lymphocytes and M1 macrophages are then recruited and promote inflammation. During the angiogenesis and proliferation stage, fibroblasts migrate to the wound and macrophages polarization is modified. Hypoxia is reduced and ROS level decreases indicating the beginning of the late phase. During the remodeling phase, macrophages are polarized in an M2 resolving phenotype and fibroblasts are still present. ROS return to a physiological low level and hypoxia is abolished.

Disturbance at any point in the wound healing process can contribute to pathological wound like fibrosis or non-healing wound.

### 2.1. Macrophages in Wound Healing Process

Macrophages are major contributing cells in the wound healing process following organ damage either induced by infection, autoimmune disorders, mechanical or toxic injuries. Evidence demonstrates that macrophages depletion reduces inflammatory responses whereas macrophages activation reduces recovery responses [13,17]. Beside tissue resident macrophages, bone marrow-derived macrophages, along with neutrophils, are among the first cells recruited to the site of injury. Their role, widely reviewed within the past years, is described at each step of tissue repair allowing them to be grouped into three types of activation. Firstly, early research highlighted their pro-inflammatory and scavenging contribution to the inflammatory stage [18,19]. Cellular response is then initiated by secreted inflammatory mediators (chemokines, ROS, matrix metalloproteases) leading to pathogens killing and phagocytosis [20,21]. At this stage, macrophages are mainly described with a pro-inflammatory ‘classical’ M1 phenotype. Secondly, in response to microenvironment stimuli, the predominant macrophage population can maturate to an anti-inflammatory healing phenotype depicted to remove dead cells and dampen inflammation [22,23]. These M2 resolving macrophages promote cellular proliferation and blood vessel development through growth factors (Platelet-Derived Growth Factor [PDGF], insulin-like growth factor-1, Vascular Endothelial Growth Factor [VEGF]) and reduce local hypoxia following injury [24,25]. They secrete Transforming Growth Factor-β1 (TGF-β1), which will allow fibroblasts differentiation, stromal cells migration and expansion, wound contraction and closure. In the final stage, a specific subtype of macrophages, called tissue-remodeling macrophages, instruct tissue repair suppressing immune response and subsequently resolving inflammation.

These three functional phenotypes involve an activation continuum that evolves, according to cellular ontogenesis and environmental stimuli, from a pro-inflammatory to a remodeling phenotype [26,27]. Each stage of wound healing must be carefully regulated, especially by different macrophage phenotypes whose roles are unique and critical [28].

### 2.2. ROS in Wound Healing

ROS (superoxide anion [O_2_^•–^] and hydrogen peroxide [H_2_O_2_]) act in the early phase of wound healing to induce vasoconstriction, platelet activation and defend host from bacterial invasion [11,29,30]. They play a pivotal role in orchestrating wound healing owing to the function of their signaling mediators in immune and stromal cells. ROS allow the recruitment of neutrophils, macrophages and/or lymphocytes to the site of injury [31] and promotes endothelial migration and division. Oxidative stress indicators include glutathione oxidation, modulation of redox-sensitive kinases, or transcription factors such as Nuclear Factor-Kappa B (NF-κB) [32].

ROS level is finely controlled by small anti-oxidant molecules (vitamin C, vitamin E, α-tocopherol, Nicotinamide adenine dinucleotide phosphate [NADPH]) or by an endogenous anti-oxidant and pro-oxidant specialized group of enzymes [33]. Anti-oxidant enzymes (catalase [CAT], glutathione peroxidase [GPx], superoxide dismutase [SOD], NADPH quinone oxidoreductase-1 [NQO-1], Heme-oxygenase-1 [HO-1]) are designed to detoxify ROS and thereby eliminate their deleterious effects. Contrariwise, NADPH oxidases (NOXs) are a family of major ROS-producing enzymes. The seven transmembrane isoforms (NOX1, NOX2, NOX3, NOX4, NOX5, Duox1, and Duox2) have tissue- and cell type-specific expression profiles and are involved in ROS production as NOX2 and NOX4 mRNA are overexpressed in injury [34]. In addition to the control of the redox state, these enzymes are implicated in a wide range of cellular processes, which includes apoptosis, cellular signal transduction, host defense, angiogenesis and oxygen sensing [35].

A precise homeostatic control of oxidative state is essential for normal tissue repair while extreme (low or high) levels of ROS can impair wound healing [36,37,38,39]. Indeed, several studies indicated that reduced ROS level, by magnetic field or pro-oxidant enzyme deficiency, improved wound healing in a model of diabetic mice [34,40]. Furthermore, it has been well documented that non-healing wounds, due to diabetes, or chronic wounds characteristic of pathologies such as inflammatory bowel diseases, are associated with a higher ROS level [32,41,42,43]. Elevated and sustained ROS are, in these cases, due to excessive or uncontrolled oxidant production or decreased anti-oxidants level (Vitamin E, glutathione) or enzymes activity (CAT, GPx or SOD) [44,45]. This results in a prolonged inflammation process. It therefore appears important to be able to modulate ROS production in healing and to redirect the therapeutic strategy towards their control.

## 3. Macrophages Polarization: Role of ROS and NOXs

Because macrophages and ROS play major roles in the process of tissue repair, we will focus this review on the mechanisms underlying ROS production during macrophages differentiation and polarization in an oxidative microenvironment.

Based on hydroxyl radical (HO^●^) imaging, macrophages differentiation stage and HO^●^ formation are closely interlinked and involve NADPH and consequently NOXs [46]. Furthermore, macrophages polarization towards the pro-inflammatory M1 phenotype resulted in an increased O_2_^•–^ and H_2_O_2_ production compared to M2-polarized macrophages [47]. These results suggest the implication of NOX enzymes in this process.

Further evidence identified the pro-oxidants enzymes NOX1, NOX2 and NOX4 in phagocytes [47,48]. NOX1 and NOX2 are the main isotypes expressed in both bone marrow monocytes and bone marrow-derived macrophages [49]. NOX2 is the most well-characterized enzyme for its role in phagocytic function and is the highest expressed in both human and murine immature macrophages, followed by NOX4 and NOX1 [47,49].

### 3.1. ROS in Macrophages Differentiation/Polarization and Function

Macrophages produce ROS, which can modulate macrophages function at various stages. Firstly, ROS are essential for the monocytes to macrophages differentiation. Indeed, previous studies indicated that chemically inhibition of ROS generation may affect the monocyte-macrophage differentiation process. Treatment with butylated hydroxyanisole (BHA), a ROS inhibitor, during differentiation blocked the increase in the expression of the macrophage marker CD11b, the induction of O_2_^•–^ production and the specific macrophage morphology features [50,51]. This loss of morphology was partially recovered by low concentrations of H_2_O_2_ [50]. In the context of healing, ROS produced by neutrophils allow bone marrow monocytes to differentiate into macrophages [52].

Secondly, ROS are required for M2 differentiation. ROS inhibitors have been reported to block the overexpression of the M2 marker CD163, the M2 cytokine interleukin-10 (IL-10) and the chemokines CCL17, CCL18 and CCL24 [50,51]. ROS inhibition only acts during the polarization stage and has no effect on the phenotype and function once the macrophage is mature. Indeed, decreased M2 ROS production, after lipopolysaccharide (LPS) treatment, do not affect the expression of M2 markers such as CD163 or CD200R [53]. With regard to the pro-inflammatory macrophages, this treatment had no effect on the CD86 marker and little effect on the secretion of M1 cytokines, Tumor Necrosis Factor-α (TNF-α) and IL-6 [50]. Other studies concluded that depletion of H_2_O_2_ by catalase or inhibition of ROS favors the expression of M1 markers on bone marrow-derived macrophages [47] and a function on T cell proliferation comparable to M1 macrophages [51]. It therefore seems that ROS are required for macrophages differentiation and polarization.

### 3.2. NOXs in Macrophages Polarization

At the molecular level, NOX1 and NOX2 are implicated in this process. Indeed, in monocytes from NOX1/2 double knockout mice, ROS generation was largely blocked and affected macrophages differentiation resulting in more rounded and less differentiated cells [49]. NOX2 and its product O_2_^•–^ specifically promote an M1 phenotype with phagocytic activity and pro-inflammatory properties [54,55]. Accordingly, NOX2 deficiency reduced pro-inflammatory M1 macrophages and promoted M2 macrophages polarization in a mouse model of brain injury [56]. In contrast, M2 polarization of macrophages is characterized by both reduced NOX2 activity and reduced O_2_^•–^ production. Loss of NOX1 and NOX2 affects the differentiation of monocytes to macrophages and the polarization of M2 macrophages. The M2 populations from NOX1/2 double knockout mice were substantially reduced compared with the wild-type mice [49]. In a wound healing model, NOX1/2 double knockout mice had less infiltration of M2-type macrophages in the wound edge and a delayed wound healing compared with wild-type mice [49,57]. These results may indicate a defect in macrophages polarization or recruitment to the site of injury. Another study, on in vitro murine macrophages, indicated that loss of NOX2 induced a small but significant reduction in M1 polarization with no effect on M2 polarization [47,49].

Because NOX4 expression is increased during phorbol myristate acetate (PMA)-induced monocytes to macrophages differentiation, several studies analyzed the contribution of this enzyme on this process. Data showed that NOX4 expression remained upregulated in the PMA-induced differentiating macrophages, while treatment with apocynin downregulated NOX4 in an in vitro system [46]. When NOX4 was chemically inhibited, TNF-α and IL-1β expression was increased in human macrophages, derived from peripheral blood monocytes, indicating M1 polarization. This was accompanied by a significant downregulation in M2 markers [47]. On the contrary, other studies focused on murine intestinal macrophages abundantly found in inflammatory bowel diseases and expressing various phenotypes. They revealed that NOX4 inhibitor suppressed the M1 polarization of intestinal macrophages, reducing the proportion of F4/80^+^ CD11c^+^ macrophages and inflammatory cytokines levels [58]. We can assume that these divergent results of NOX4 inhibition relate with the macrophages lineage and that NOX4 may act on distinct differentiation and polarization stages. Furthermore, the absence of NOX4 increased ROS formation in M1-polarized macrophages. Because the major source of ROS in M1 macrophages is NOX2, studies revealed that its expression was elevated in NOX4-deficient M1 polarized macrophages [47].

### 3.3. Molecular Events and Signaling Pathways Involved in Wound Healing

Wound healing stages involve specific molecular hallmarks such as hypoxia, inflammation and oxidative stress. These markers are regulated, among other things, by numerous transcription factors. Activation of these transcription factors is a key event for the hypoxic or inflammatory signaling cascades and the oxidative stress response. We will describe here the main signaling targets identified in macrophages (i.e., Hypoxia Induced Factor [HIF], NF-κB and nuclear factor erythroid-2-related factor 2 [Nrf2]) and their functional interrelation (Figure 2).

-HIF (Figure 2 ①)

In a wound, local oxygen level is reduced due to blood vessel destruction [59]. A change in oxygen concentration regulates transcription factors, the main being HIF. This local hypoxia implicates macrophages and induces ROS production, among others, as signaling molecules to restore normoxia [42,60]. As oxidative stress and macrophages are closely related during wound healing, the role of ROS on HIF activation, in macrophages, has been investigated.

HIF are a family of 3 transcription factors (HIF-1, HIF-2 and HIF-3). These heterodimers of β-subunits and hypoxia-induced α-subunits (HIF-1α, HIF-2α HIF-3α bind to hypoxia-responsive elements and activate target genes transcription. HIF-1α induces the expression of glucose transporter 1 (GLUT1), and pyruvate dehydrogenase kinase isoform 1 (PDK1) in macrophages [61,62]. In cancer cells, HIF-1α increases Programmed death-ligand 1 (PD-L1) expression and cytokines secretion (i.e., VEGF) thereby promoting tumor associated macrophages (TAM) accumulation and immune escape.

In homeostatic conditions, HIF activation is regulated by proteasomal degradation. HIF is hydroxylated by prolyl hydroxylases (PHDs) and subsequently ubiquitinated by the E3 ubiquitin ligase von Hippel-Lindau. These modifications direct HIF to the ubiquitin-proteasome system for degradation. Another layer of regulation involves the interaction between proteins from the signaling pathway. This level involves Factor inhibiting HIF (FIH), which blocks interactions between the HIF-α transactivation domain and coactivators. When oxygen concentration decreases, PHDs are inactive and HIF is stabilized in the cytoplasm. This accumulation allows the transcription factor to translocate in the nucleus and to regulate target genes expression [63].

Several studies have focused on oxidative stress and HIF during hypoxia or normoxia [64,65]. They revealed that ROS contribute to HIF transcriptional activity by stabilizing HIF-1α and inhibiting FIH. Indeed, Chandel et al. demonstrate that catalase abolishes HIF-1α stabilization under hypoxic conditions [64]. Conversely, high concentration of H_2_O_2_ can induce HIF-1α stabilization in normoxia [65].

In macrophages, stimuli like LPS or pathogenic microorganisms’ infection, can upregulate HIF-1α expression and activity through NF-κB signaling [65,66,67]. Indeed, Li et al. demonstrate the critical role of HIF-1α during macrophages polarization towards pro-inflammatory phenotype. They also found that HIF-1α is necessary for macrophages responses when these cells are challenged with pathogens. Furthermore, in HIF-1α deficient macrophages, mRNA expression, production and secretion of several pro-inflammatory cytokines (TNF-α and IL-6) or VEGF are inhibited independently of oxygen level [67,68]. In inflammatory bowel diseases, on the contrary, effects of HIF knockout in myeloid cells depend on the type of transcription factor studied. Finally, in an intestinal context, HIF-1 has been reported to promote inflammation while HIF-2 protects against chemically induced inflammation [69].

-NF-κB (Figure 2 ②)

NF-κB plays a crucial role in inflammatory and immune responses and is subject to complex regulation. It participates in a plethora of macrophages regulatory mechanisms and is associated with extensive ROS production. Its role in healing is therefore important at all stages of the process, whether it is at the early inflammatory phase or at the later phase of tissue formation and remodeling [70,71].

NF-κB is a homo- and hetero-dimeric complex resulting from the five monomers in mammals (RelA/p65, RelB, cRel, NF-κB1 p50, and NF-κB2 p52) [72]. The heterogeneity of NF-κB targets is further increased by interactions of NF-κB dimers with other transcription factors. The most well characterized heterodimer during inflammatory response is the p50/p65 complex. NF-κB is kept inactive in the cytosol by binding to the inhibitory protein IκBα (nuclear factor of kappa light polypeptide gene enhancer in B-cells inhibitor, alpha). Under various stimuli (inflammation, cytosolic ROS), the IκB kinase (IKK) complex, which is constituted of two catalytic subunits IKKα and IKKβ and a regulatory subunit IKKγ (or NEMO), is phosphorylated. This complex, thus activated, phosphorylates IκBα thereby targeting the protein for proteasomal degradation. NF-κB is then free to translocate to the nucleus and initiate the transcription of several genes [70,71,73]. ROS can also oxidize NF-κB cysteines and inhibit its DNA binding, reducing its activity. In addition to its major role in inflammation, an immunosuppressive one has been described in the context of tumor microenvironment where ROS induce PD-L1 expression through NF-κB binding to its promoter [74]. In the same way, in an inflammatory bowel disease model, ROS activate NF-κB signaling leading to the recruitment and the polarization of intestinal macrophages to an M2 phenotype [75]. In metabolic disorders such as obesity and type 2 diabetes, Luo et al. demonstrate that celastrol, a natural anti-oxidant, is able to suppress M1 macrophage polarization and enhance M2 polarization through inhibition of NF-κB nuclear translocation. This M1 polarization is mediated by the Nrf2 activation pathway [76].

-Nrf2 (Figure 2 ③)

Transcription factor Nrf2 is a basic leucine zipper (bZIP), which is a major sensor for oxidative stress [77]. It has been described to maintain redox homeostasis and to attenuate inflammation and thereby to be involved in wound healing [78].

Under unstressed conditions, Nrf2 is retained in the cytoplasm by Kelch-like ECH-associated protein 1 (Keap1) that functions as an Nrf2 Inhibitor [79]. Keap1 is an adapter protein of the E3 ubiquitin ligase Cul3-Ring-box 1, which is responsible for the ubiquitination and proteasomal degradation of Nrf2.

Upon oxidative stress, several cysteine residues on Keap1 are subjected to oxidation which induced a conformational change in the protein and prevents Nrf2 ubiquitination and subsequent degradation. As a consequence, Nrf2 is released from Keap1 and accumulates in the cytoplasm. Nrf2 then translocates into the nucleus and forms a heterodimer with bZIP proteins. On one hand, the heterodimer Nrf2 binds to anti-oxidant response elements of target genes and regulates the expression of cytoprotective anti-oxidant genes and detoxifying enzymes implicated in NADPH, glutathione and thioredoxin systems (HO-1 and NQO-1) [80]. Nrf2 is also implicated in NOX expression as its deletion in fibroblast induces an upregulation of NOX4 [81]. As Nrf2 is essential to maintain redox homeostasis, its inhibition in fibroblasts reduces specific NADPH ROS production during treatment with ionomycin (a calcium ionophore agent) while it does not interfere with ROS levels in basal conditions. In Nrf2 knockout mice, ROS level is increased compared to wild-type mice [81].

On the other hand, Nrf2 is described to regulate gene expression of pro-inflammatory cytokines independently of ROS level [82]. In this case, evidence suggested that Nrf2 can bind to the proximity of the pro-inflammatory gene (not only on anti-oxidant response elements) and interferes with the polymerase II thereby inhibiting the transcription initiation step [82].

Furthermore, in macrophages, a high level of Nrf2 decreases LPS-induced cytokines while, in its absence, pro-inflammatory cytokines are upregulated [83,84]. Microarrays analyses, on bone marrow derived macrophages from Nrf2 knockout mice, indicated that genes induced during M1 polarization are downregulated [82,83]. Other indirect evidence suggests that Nrf2 induces M2 macrophages polarization. Overexpression of HO-1, a Nrf2 target gene, induces an anti-inflammatory response in cultured macrophages [85]. In a model of delayed diabetic wound healing, Nrf2 activation accelerates the wound process while Nrf2 inhibition mimics the effects of diabetes and the delayed process [84]. In inflammatory bowel diseases, Nrf2 has been reported to protect against colitis. The first study describing this role, performed by Khor et al., reveals that Nrf2 knockout mice are more sensitive to chemically induced colitis [86]. Further studies indicate that Nrf2 prevents the early stages of carcinogenesis associated with colitis [87].

Nevertheless, Nrf2 has been also described to favor the progression of cancer cells. In TAM, nuclear translocation of Nrf2 is increased and its targeted anti-oxidant genes are overexpressed. In Nrf2 knockdown macrophages, treatment with cancer cell medium blocked the induced over-expression of M2 markers and down-regulation of M1 markers [88]. Controversially, in macrophages exposed to the tumor fluid, data indicate that Nrf2 nuclear localization is reduced, indicating an alteration in the oxidative status [89].

In wound healing, signaling pathways are closely linked and are activated at different stages of the process. Their activation is not stage-specific but presents a continuum. Their roles are in some cases redundant and allow the activation of the same target genes, therefore having an identical overall effect. In some cases, transcription of target genes from one signaling pathway will activate another pathway. It is therefore difficult to know exactly the role of each signaling pathway in wound healing.

## 4. Metastasis

Many wound healing cellular actors, molecular mechanisms and signaling pathways are also implicated in metastasis [2]. Therefore, elucidating the link between wound healing and metastatic cancer progression may allow the development of better therapeutic strategies against these two pathologies.

### 4.1. Metastasis Hallmarks

Metastatic spread comprises a complex succession of cellular biological events leading to the dissemination of cancer cells from the tumor to the surrounding tissues and to distant organs, through blood and lymphatic vessels [90]. Furthermore, it also involves crosstalk between cancer cells and components of the tumor microenvironment [91].

The metastatic process begins with the hypoxia at the primary tumor site due to excessive cell proliferation [92]. Reduced oxygen level induces HIF-1α stabilization and its nuclear translocation, which promotes the expression of various genes involved among others in angiogenesis, glucose metabolism, extracellular matrix remodeling, epithelial-mesenchymal transition, metastasis, cancer stem cell maintenance and immune invasion [93]. In parallel, hypoxia-induced necrosis results in a continuous release of cellular debris, notably High Mobility Group Box protein-1 (HMGB1) by dying tumor cells [94]. HMGB1 has been shown to be up-regulated in tissue biopsies from cancer patients [95]. Interestingly, HMGB1 plays opposite roles depending on its redox state. Oxidized HMGB1 induces the production of pro-inflammatory cytokines whereas the reduced form interacts with TAM therefore regulating monocyte recruitment, angiogenesis and immune suppression [96]. In fine, altering the redox status of HMGB1 may be considered as a therapeutic approach to combat metastasis and favor wound healing.

Angiogenesis provides oxygen and nutrients supply essential for cancer cells to dissociate from the basal membrane delineating the epithelial compartment from the stroma. This requires the degradation of the extracellular matrix (ECM), through the activation of matrix metalloproteinases [97]. Under normal circumstances, cells detachment from the ECM leads to the induction of an apoptosis called anoikis, a form of programmed cell death that occurs in anchorage-dependent cells [98]. However, cancer cells develop a trans-differentiation program known as epithelial–mesenchymal transition (EMT), which render the cells resistant to anoikis [99]. Anoikis plays an important role in the prevention of metastasis and promoting its induction might be an interesting therapeutic strategy. Finally, cells acquire stemness properties. Stemness is the ability of a cell to perform self-renewal and is capable of pluripotency. This is an important feature for supplying material for wound closure and for the establishment of cancer cells at the metastatic sites [100].

### 4.2. ROS and Metastasis

One of the principal mechanisms underlying metastasis in human cells is the disruption of the redox balance. This imbalance in redox homeostasis is induced by an increase in free radicals, mainly ROS [101]. Cancer cells have elevated expression levels of NOXs (NOX1, NOX2, NOX4, NOX5), leading to high levels of ROS [101,102]. Consequently, cancer cells have been shown to be more tolerant to oxidative stress via increased expression of catalase and superoxide dismutase. However, the lack of robust anti-oxidant defenses may have detrimental consequences in the tumor microenvironment and in the adjacent normal cells [103].

-Dual effect of ROS

Although several processes of metastasis are redox-sensitive, it is still controversial whether ROS have oncogenic/metastatic or tumor suppressive functions. The answer appears to depend on ROS levels and the cancer stage, leading many authors to consider ROS as a “double-edged sword” [101]. Low to moderate ROS levels can promote survival of cancer cells by inducing EMT and stem cell differentiation, enhancing angiogenesis and switching to glycolytic metabolism. Conversely, excessive production of ROS induced by chemotherapy and radiotherapy is detrimental to the survival of cancer cells and causes cellular damage [104,105]. Concerning the stage of the disease, it has been reported that in the early stages of cancer, ROS promote cancer initiation by inducing base pair substitution mutations in pro-oncogenes such as Ras and tumor suppressor genes such as p53 [106]. As cancer progresses, an intracellular excess of ROS triggers apoptosis of tumor cells. To escape this ROS-induced apoptosis, tumor cells produce high levels of anti-oxidants [106]. In the last stages of tumor development, ROS have a pro-metastatic role promoting the spread of cancer cells.

-ROS and angiogenesis

Additionally, ROS are involved in angiogenesis. Angiogenesis is mainly mediated by VEGF whose expression can be regulated by nutrient deprivation and hypoxia, both of which increase levels of ROS [107,108]. Activation of angiogenesis by ROS can involve different signaling pathways. Firstly, ROS have been shown to activate PI3K/Akt/mTOR signaling cascade in different cancer cell lines (MCF-7, HepG2, H-1299, PC-3), enhancing HIF-1α and VEGF expression and ultimately angiogenesis [109,110]. The role of ROS has been confirmed by several studies showing that catalase and glutathione peroxidase overexpression or NOX4 knockdown lead to a decrease in VEGF and HIF-1α levels and inhibit angiogenesis in human ovarian cancer cells [111,112]. Further, oxidative stress can induce angiogenesis in a VEGF-independent manner through the activation of the TLR/NF-κB pathway. West et al. demonstrated the proangiogenic effects of TLR1/2 stimulation by oxidative stress, represented by lipid oxidation products, in murine and human melanoma [113]. In addition, angiogenesis is also mediated by matrix metalloproteinases and upregulated by ROS [114].

-ROS, EMT and anoikis resistance

Several studies have proven that ROS are a major cause of EMT. ROS-induced EMT has been reported to be NOX4-dependent in human metastatic breast epithelial cells [115] and in lung cancer cells [116]. NOX4 is an important source of ROS induced by TGF-β and under hypoxia, two important mediators in cancer metastasis [117,118]. Furthermore, NOX4 inhibition significantly attenuated the distant metastasis of breast cancer cells to lung and bone [119].

Resistance to anoikis seems to concern not only the field of cancer but also this phenomenon may be interesting in wound healing. Indeed, ROS are considered as one of the key players in anoikis sensitivity. In recent studies, ROS generation induced by NOX4 has been involved in anoikis resistance of gastric [120] and lung cancer cells [121]. ROS promote EMT by inducing the expression and activity of MMPs that mediate proteolytic degradation of ECM components [122,123]. TGF-β1, a well-established player of EMT induction, regulates MMP-9 to facilitate cell migration and invasion via the activation of NF-κB through a ROS-dependent mechanism [123]. Similarly, ROS production induced MMP-2 secretion and activation results in pancreatic cells invasion [122]. In colorectal cancer, the EMT process is highly regulated through some of the classic tumorigenic signaling pathways, such as the NF-κB, HIF-1, and TGF-β1 pathways [124]. Intriguingly, TGF-β1 induces EMT through Nrf2 activation as well as ROS production in lung adenocarcinoma cells [116]. Indeed, Nrf2 is a key transcriptional regulator that drives anti-oxidant gene expression and protection from oxidative damages. Oxidative stress plays a critical regulatory role in these pathways by degrading inhibitors or inducing nuclear translocation and consequent transcription [124].

-ROS and stemness

Cancer stem cells possess a particular redox status, since they have lower ROS levels and increased anti-oxidant capacity than differentiated cancer cells [125,126]. Increasing evidence shows that these low amounts of ROS are actually needed to maintain the quiescence and self-renewal potential of cancer stem cells (CSC). Previous studies have demonstrated that ROS contribute to reduce stemness and to enhance differentiation of CSC. For example, glioblastoma stem cells have potent anti-oxidant defense mechanisms and H_2_O_2_ has been shown to inhibit their self-renewal and induce their differentiation [127]. ROS have been reported to promote hematopoietic stem cell differentiation with a progressive increase in ROS levels with the advancing differentiation stages. Moreover, inhibition of ROS production has been found to attenuate the differentiation of hematopoietic stem cells [128]. In summary, hypoxia-associated increase in ROS in tumor cells promotes stemness. Although oxidative stress promotes the development of CSC, ROS level declines after this acquisition of stemness, allowing the maintenance of the sub-population.

### 4.3. Oxidative Stress and Metastasis: Cellular Actors Involved

Macrophages, neutrophils and fibroblasts are major ROS producers in the tumor microenvironment [92]. Here, we will focus on macrophages and fibroblasts since neutrophils activation in wound healing and metastasis has been already extensively reviewed [129].

-Macrophages

In cancer, macrophages present in the tumor are known as TAM and can represent up to 50% of the tumor mass [130]. ROS can be both beneficial and detrimental for the anti-cancer immune function. Therefore, they may indirectly impact cancer progression by altering cancer immune surveillance [131]. Although macrophages have anti-tumor effects as immune cells, experimental and clinical evidence have revealed that TAM contribute to tumor progression and metastasis. High levels of TAM are associated with weak prognosis and decreased overall survival in various cancers [132,133,134,135]. The effect of ROS in TAM polarization toward a M1 or M2 phenotype has been discussed, as several studies showed that ROS can stimulate both activation statuses in TAM [49,50,136,137]. M1 and M2 macrophages are two extremes in a continuum of macrophage functional states, which reflect the different effects that can be observed on tumor cells [138].

O_2_^•–^ production promotes M2 polarization through activation of ERK and JNK signaling pathways [49,50]. Moreover, administration of the anti-oxidant BHA blocked TAM infiltration and tumor progression, which suggests a beneficial effect of ROS inhibition in tumor therapy [50]. Indeed, another ROS scavenger, oligo-fucoidan, has been reported to inhibit M2 polarization and TAM infiltration in subcutaneous colorectal tumors [139]. Conversely, Wu et al. demonstrated that increased NOX-dependent ROS production by irradiation of macrophages promotes a pro-inflammatory M1 phenotype that is associated with improved response to radiotherapy in rectal cancer [137]. Similarly, iron overload has been reported to polarize macrophages towards an M1 phenotype by increasing ROS production and reduction in ROS levels by N-Acetyl-Cysteine repressed M1 polarization [136]. These results confirm a link between ROS generation and M1 polarization of macrophages. Apart from polarization, ROS also govern TAM apoptosis. For example, inhibition of autophagy in macrophages increases ROS levels, provokes TAM apoptosis and leads to regression of the primary tumor [140]. TAM are also major players in the regulation of tumor angiogenesis in colorectal cancer [141]. They have been demonstrated to enhance the expression of angiogenic proteins in the tumor microenvironment in an oxidative stress-dependent manner by regulating the activity of NOXs [142].

-Fibroblasts

In wound healing, fibroblast’s function includes renewal of ECM, the regulation of epithelial differentiation and the regulation of inflammation. Cancer-Associated Fibroblasts (CAFs) are the most predominant stromal cell type in the tumor microenvironment [143]. They are major producers of ROS [144], which facilitates metastasis through the activation of angiogenesis [145]. Moreover, cancer cells induce ROS overproduction in CAFs contributing to a pro-oxidative tumor microenvironment [146]. Conversely, ROS produced by CAFs enhance ROS generation in cancer cells, increasing tumor aggressiveness [147]. CAF-mediated ROS production are involved in the increased metastasis potential of prostate carcinoma. CAF drive cancer cells to secrete cyclooxygenase-2 (COX-2)-mediated ROS, which is mandatory for EMT, stemness and dissemination of metastatic cells [148]. Finally, CAFs, in a mouse model of squamous skin carcinogenesis, promote macrophage recruitment and neovascularization in close association with NF-κB [149].

## 5. Conclusion and Future Perspectives

Although the intertwining of wound healing and metastasis have already been well described in the literature, this review highlights the molecular and cellular similarities between these two processes. Notably, accumulating evidence designates ROS and macrophages as major regulators of these pathologies, in which disturbance can lead to either pathological wounds or cancer cells spread. These two actors are intrinsically linked since macrophages are the main source of oxidative stress and, at the same time, their differentiation and polarization require ROS. In this context, both appear as potential therapeutic targets.

As recapitulated in Figure 3, a high level of ROS is a common feature in the development of non-healing wound and metastasis. Controlling oxidative stress level in wound and tumor cells environment can be an interesting strategy both to promote wound healing and to prevent metastatic spread. The excessive ROS accumulation could be managed by (1) scavenging agents, (2) limiting its production and/or (3) increasing anti-oxidant defenses. ROS-scavenging hydrogel showed enhanced wound healing abilities by down-regulating pro-inflammatory cytokines, up-regulating the M2 phenotype of macrophages and promoting angiogenesis and the production of collagen [150]. Secondly, the production of ROS can be limited through NOXs inhibition. To date, few studies have focused on this area due to the lack of specificity and pharmacological knowledge on NOXs inhibitors [151]. Nevertheless, a dual protective effect against oxidative stress has been demonstrated by beta3-adrenergic receptor stimulation on macrophages. Indeed, it results in the inhibition of NOXs activity, a decreased NOX2 level and an increased catalase expression [152]. Although this study was conducted for preterm birth management, the use of beta3-adrenergic receptor agonists can be applied to other pathologies associated with excessive oxidative stress production. Finally, the use of anti-oxidants such as vitamins, polyphenols and flavonoids has been widely studied [102,153]. Unfortunately, when used as monotherapy, clinical studies did not provide any therapeutic benefit. Along with the tremendous rise of the immune-checkpoint modulators as anti-cancer drugs, this led researchers to investigate the potential synergistic effects of ROS blockade and immunotherapy. For example, recent studies reported that vitamin C supplementation improved anti-cancer immunotherapies efficiency in various murine tumor models [154,155].

Reprogramming of macrophages appears as the second target for the management of cancer metastasis and, by extension, of wound healing. Indeed, since macrophages are also involved in wound pathophysiology, this therapeutic approach can also be interesting in wound healing. Administration of the anti-oxidant BHA blocked M2 macrophage differentiation resulting in suppression of tumorigenesis in three different mouse cancer models [50]. Similarly, another ROS scavenger, oligo-fucoidan, induced monocyte polarization toward M1-like macrophages and repolarized M2 macrophages into M1 phenotypes; therefore, inhibiting colorectal tumor progression [139].

It is worth mentioning that some limitations of targeting oxidative stress as a promising treatment in wound healing and metastasis relies on the balance needed between beneficial and harmful effects of ROS. As a double-faceted agent, ROS also play a pivotal role in orchestrating wound healing mechanisms [156] and as potent genotoxic agents causing DNA damage in cancer cells [102]. As proof, radiotherapy and chemotherapy induce oxidative stress necessary for their anti-tumoral activity [104,105]. Furthermore, due to some disparities in the mechanisms of these two diseases, questions arise as to the modalities and timing of administration of therapies. Defective wound healing would require local treatment while systemic treatment seems more suitable to prevent and treat metastases.

In summary, this review offers a compilation that may provide a better understanding of the pivotal role of oxidative stress in both wound healing and metastasis, encompassing the contribution of macrophages. Although the treatment of metastases or chronic wounds is a real challenge, new therapeutic approaches involving administration of redox modulators need to be considered. 

## 6. Methods

This literature review was based on searches on PubMed, Web of science, Springer and Wiley databases, with no time limit but giving preference to recent articles.

## Figures and Tables

**Figure 1 biomedicines-10-02784-f001:**
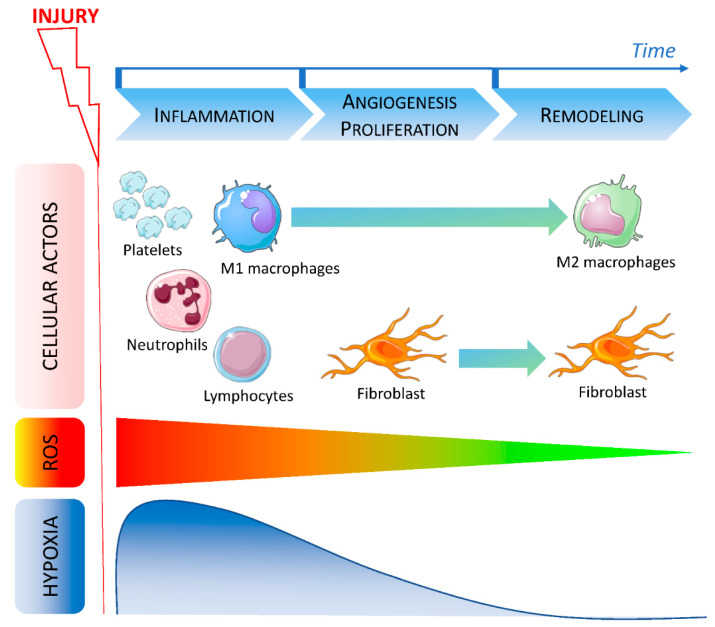
Timeline of cellular actors, ROS and hypoxia involved in wound healing.

**Figure 2 biomedicines-10-02784-f002:**
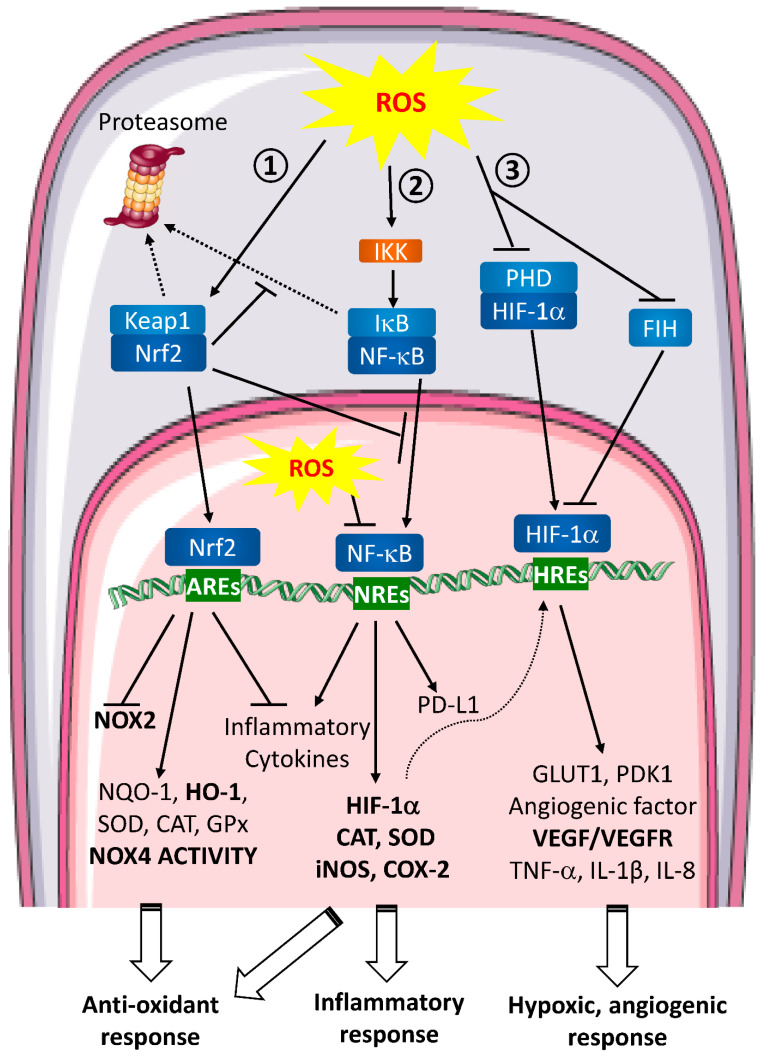
ROS signaling pathways involved in wound healing and metastasis. Extracellular ROS activate intracellular signaling pathways. ① Nrf2 pathway. In unstressed conditions, Keap1 retains Nrf2 in the cytoplasm. When ROS are produced, Keap1 is oxidized and ubiquitinated thereby leading to its proteasomal degradation. Consequently, Nrf2 is free to translocate to the nucleus and binds to the anti-oxidant response elements (AREs). This binding inhibits NOX2 and pro-inflammatory cytokines transcription and enhances the anti-oxidant defense response expression. ② NF-κB pathway. In normal conditions, NF-κB is associated with IκB and retained in the cytoplasm. In the presence of ROS, IKK is activated and can phosphorylate IκB to induce its dissociation with NF-κB and its proteasomal degradation. Then, free NF-κB translocates to the nucleus, binds to NF-κB Response Elements (NREs) and induces target genes transcription leading to a global inflammatory response. ROS are able to directly act in the nucleus inhibiting NF-κB binding to the NREs. ③ HIF pathway. In homeostatic conditions, HIF-1α is hydroxylated by PHDs and targeted for proteasomal degradation. HIF-1α is also regulated by FIH, which blocks the interaction between HIF-1α transactivation domain and coactivators on HREs. During hypoxia or when ROS are produced, PHDs are inactivated which stabilizes HIF-1α and FIH is inhibited. HIF-1α then translocates into the nucleus where it binds to HIF Response Elements (HREs). Transcription of target genes is induced leading to hypoxic and angiogenic response. Nrf2, NF-κB and HIF pathways are closely interlinked. Nrf2 can inhibit IκB proteasomal degradation and NF-κB nuclear translocation, NF-κB pathway induces anti-oxidant response regulating iNOS and COX-2 transcription and HIF-1α expression is regulated by NF-κB.

**Figure 3 biomedicines-10-02784-f003:**
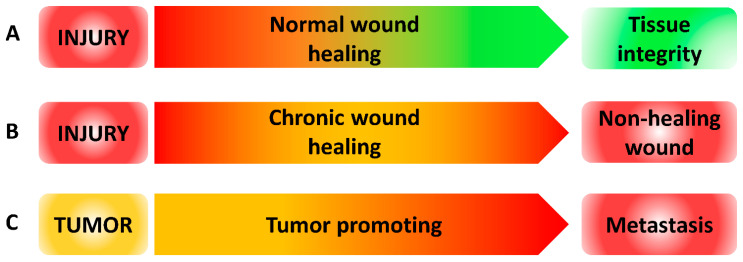
ROS levels during wound healing and metastasis. (**A**) ROS level during normal wound healing. After injury, high levels of ROS (red) are produced and then decreased to low level (green) over time to restore tissue integrity. (**B**) ROS level during chronic wound healing. After injury, high levels of ROS (red) are produced and failed to be reduced inducing non-healing wound. (**C**) ROS level during tumor progression. In tumor, ROS are produced in an intermediate level (orange). When ROS level increased, tumor progression is promoted leading to metastasis.

## Data Availability

The data that support the findings of this study are available from the corresponding author upon reasonable request.

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
