# Peer review of "Wound Healing versus Metastasis: Role of Oxidative Stress"

_biomedicines, 2022, doi:10.3390/biomedicines10112784_

Round 1

Reviewer 1 Report

In this review, the Authors describe some molecular mechanisms in common with wound healing and metastases related to the presence of ROS. 

The manuscript is fascinating and well organised. However, I have some questions and points that I would like to bring to the authors' attention.

1. Authors should recheck the font and font size of the entire manuscript and always use the same.

2. Figures 1 and 3 are unclear. Authors speak of "timeline", but the figures do not mention time.

3. I would not put the acronyms in the list of keywords.

4. The manuscript is full of acronyms. Perhaps it would be helpful to make a list of abbreviations and their extended form to be inserted at the beginning or end of the manuscript, if the Journal allows it.

5. Just a curiosity: what do the authors think of the role of mast cells and myofibroblasts? in the manuscript, they are never mentioned, even in the figures. Despite this, mast cells release histamine, which plays an essential role in the early stages of inflammation and, thus, in wound healing. 

6. I think the Authors should improve Figure 3. They link the chronic wound to fibrosis. However, by definition, the chronic wound does not heal but remains stationary at the inflammatory stage. On the contrary, fibrosis is an excess of healing. 

Author Response

Responses to Reviewer 1 concerning the manuscript:
Manuscript ID: biomedicines-1953191
Title: Wound healing versus metastasis: role of oxidative stress
Authors: Tatiana Lopez, Maeva Wendremaire, Jimmy Lagarde, Oriane Duquet, Line Alibert, Brice Paquette, Carmen Garrido, Frédéric Lirussi *
We thank the Reviewer for his/her positive comments on our manuscript. In particular, we appreciate the Reviewer’s recognition that this work is fascinating. We considered and answered his comments.
In this review, the Authors describe some molecular mechanisms in common with wound healing and metastases related to the presence of ROS.
The manuscript is fascinating and well organised. However, I have some questions and points that I would like to bring to the authors' attention.
1. Authors should recheck the font and font size of the entire manuscript and always use the same.
Response 1: The font and font size have been harmonized throughout the manuscript.
2. Figures 1 and 3 are unclear. Authors speak of "timeline", but the figures do not mention time.
Response 2: We agree with the reviewer’s remark. We added a time scale on figure 1 and renamed figure 3 as “ROS levels during wound healing and metastasis”.
3. I would not put the acronyms in the list of keywords.
Response 3: We removed acronyms from the list of keywords.
4. The manuscript is full of acronyms. Perhaps it would be helpful to make a list of abbreviations and their extended form to be inserted at the beginning or end of the manuscript, if the Journal allows it.
Response 4: We agree with the reviewer’s remark and we are willing to add a table of acronyms on editor’s request.
5. Just a curiosity: what do the authors think of the role of mast cells and myofibroblasts? in the manuscript, they are never mentioned, even in the figures. Despite this, mast cells release histamine, which plays an essential role in the early stages of inflammation and, thus, in wound healing.
Response 5: This is a very interesting point that we have not mentioned. We focused on macrophages involved in wound healing and metastasis spread. Although the role of mast cells is well described in wound healing, their role on metastasis process is more intricate (Mast cells:
the Jekyll and Hyde of tumor growth - Theoharides & Conti - Trends Immunol. 2004). Moreover, macrophages are both producers and victims of oxidative stress. Lastly, we didn’t want to overload the content of the review.
6. I think the Authors should improve Figure 3. They link the chronic wound to fibrosis. However, by definition, the chronic wound does not heal but remains stationary at the inflammatory stage. On the contrary, fibrosis is an excess of healing.

Response 6: It is a relevant remark. We modified the figure 3 by replacing “fibrosis” by “non-healing wound”

Reviewer 2 Report

Pg 2, Ln 8. The statement “Three distinct and overlapping steps are needed in the physiological wound healing process…” is controversial! This is the former perspective, where coagulation (hemostasis) and inflammation are considered as a common phase. The evidence is that coagulation (alt: thrombogenesis) and inflammation are independent events (e.g., ‘sterile inflammation’). Thus, the consensus in the field is that wound healing is a cascade of four, overlapping, phases: hemostasis, inflammation, proliferation, and maturation. See Singer & Clark 1999. doi: 10.1056/NEJM199909023411006; Martin 1997. doi: 10.1126/science.276.5309.75.

Pg 2, Ln 11. What is the evidence “Neutrophils and tissue resident macrophages are the first responding cells to the wound”? (citation required)

Pg 2, Ln 12. What is the evidence “This stage also induces immune cells invasion particularly monocytes recruited from the bone marrow and differentiated into mature inflammatory macrophages (named M1)”?

Pg 2, Ln 16. What is the evidence “Resolving anti-inflammatory macrophages (named M2) contribute to remove cells and bacteria debris by efferocytosis or phagocytosis”?

Pg 2, Ln 19. What is the evidence “…tissue remodeling macrophages promote restoration of functional integrity of tissue”?

Pg 3. Fig 1. Critical cellular populations are absent from Figure 1. Cutaneous wound healing in humans is primarily orchestrated and mediated by keratinocytes, not just macrophages, lymphocytes and granulocytes! This figure is inaccurate. It is misleading.

Pg 3. Ln 1. The pathological consequences of “Deficient wound healing due to disturbance at any point in the process” is much wider than merely “pathological fibrosis”. This statement is misleading.

Pg 3. Ln 5. Why is the role of macrophages limited to “damage either induced by infection, autoimmune disorders, mechanical or toxic injuries”? As a key ‘actor’ the role of macrophages is ubiquitous to all tissue injuries.

Pg 4. Ln 1. What is the evidence “…resident macrophages, bone marrow-derived macrophages, along with neutrophils, are among the first cells recruited to the site of injury”?

Pg 4. Ln 4. What is the “early research” that “highlighted their pro-inflammatory and scavenging contribution…”? (a citation is required).

Pg 4. Ln 5. What is the evidence “Cellular response is then initiated by secreted inflammatory mediators”?

Pg 4, Ln 8. What is the “microenvironment stimuli” to which macrophages respond “to remove dead cells and dampen inflammation”? The cited evidence is from hepatocytes (a mesenchymal type). What is the evidence this mechanism is conserved in epithelia, i.e. skin cell type?

Pg 4. Ln 11. What is the evidence “M2 phenotype, promote… proliferation and blood vessel development through growth factors (PDGF, IGF-1, VEGF) and reduce local hypoxia following injury”?

Pg 4, Ln 15. What are “stromal cells”? (…presumably they are distinct from fibroblasts.)

Pg 4, Ln 16. What are “tissue-remodeling macrophages”? We’ve been introduced to M1 and M2 phenotype macrophages; now we have another macrophage phenotype!

Pg 4, Ln 19. What is the evidence “three functional phenotypes do not involve three distinct macrophage subsets but an activation continuum that evolves, according to cellular ontogenesis and environmental stimuli, from a pro-inflammatory to a resolutive phenotype”?
I suggest that the average reader will be confused by this plethora of macrophages phenotypes are but an “activation continuum” responding to “cellular ontogenesis and environmental stimuli”. I am barely keeping up, and I work in this field! As this is a review, please substantiate your interpretations with published evidence.

Pg 4, Ln 25. What is the evidence “ROS (superoxide anion [O2•–] and hydrogen peroxide [H2O2]) act in the early phase of wound healing to induce vasoconstriction, platelet activation and defend host from bacterial invasion”?

Pg 4, Ln 28. What is the evidence “ROS signaling allows the recruitment of neutrophils, macrophages and/or lymphocytes to the site of injury”?

Pg 4, Ln 33. What is the evidence “ROS level is finely controlled by small anti-oxidant molecules…”?

Pg 4, Ln 49. What is the “well documented (evidence) that non-healing wounds, due to diabetes, or chronic wounds characteristic of pathologies such as inflammatory bowel diseases, are associated with a higher ROS level”? (sic)

Pg 5. Ln 12. What is the “Further evidence…”?

Pg 5, Ln 13. What is the meaning of “…first main isotypes found…”?

Pg 5, Ln 15. having established that there are 3 phenotypic populations of macrophages, which population is referred to in “…human and murine macrophage populations…”?

Pg 5, Ln 18. What is the evidence “ROS, …can modulate macrophages function at various stages”?

Pg 5, Ln 6. What is the evidence “ROS are essential for the monocytes to macrophages translation”? What is “monocytes to macrophages translation”?

Pg 5, Ln 17. What is the evidence “chemically inhibition of ROS generation may affect the monocyte-macrophage differentiation process”? (sic)

Pg 5, Ln 22. What is the evidence “ROS produced by neutrophils allow bone marrow monocytes to differentiate into macrophages”?

Pg 5, Ln 26. The statement “ROS inhibition only acts during the differentiation stage and has no effect on the phenotype and function of mature macrophages” contradicts previous statements outlining how ROS inhibitors affect macrophage maturation/differentiation. This also contradicts the subsequent statement (Pg 5, Ln 35) “ROS are required for macrophages differentiation and polarization”.

This reader is confused.

Pg 5, Ln 36. The section “3.2. NOXs in macrophages polarization” is predicated on the assumption that murine and human monocyte /macrophage populations possess equivalent genotypes, phenotypes, functional types, and respond equivalently to ReDox-dysregulated tissue environments.

What is the evidence that supports this assumption in human skin tissue?

Pg 6, Ln 25. What is the evidence “hypoxia, inflammation and oxidative stress… are regulated by numerous transcription factors”? This is oversimplification and ignores fundamentals of chemistry.

Pg 7, Figure 2. “ROS signaling pathways involved in wound healing and metastasis.”

What is the tubular structure adjacent to (1) (top left)?

There is some confusion in Ln 5 of the legend. It is stated that Nrf2 binding to ARE binding “inhibits… anti-oxidant defense expression” yet is illustrated as enabling transcription of NQO-1, HO-1, SOD, CAT, GPx, NOX4?

Figure 2 includes many abbreviations that are not defined herein, nor in the main text.

Pg 8, Ln 8. What is the evidence “In wound, local oxygen level is reduced due to blood vessel destruction”? (sic)

It is my understanding the opposite is true! Skin tissue has an O2 pp of approx. 8.0 ±3.2 mm Hg. The O2 of air (at sea level) is 160 mm Hg. Surface oxygen is ~20 times greater, and triggers many events associated with tissue injury! I think the authors are referring to the sub-cutaneous wound bed.

Pg 8, Ln 9. What is the evidence “…local hypoxia implicates macrophages and induces ROS production”?

Pg 8, Ln 37. What is “a positive regulation of HIF-1”?

Pg 8, Ln 52. What is the evidence “NF-kB is a homo- and hetero-dimeric complex resulting from the five monomers”?

Pg 9, Ln 20. What is the evidence “Nrf2 is a basic leucine zipper (bZIP), which is a major sensor for oxidative stress”?

Pg 10, Ln 16. The references cited in support of the statement “Many wound healing cellular actors, molecular mechanisms and signaling pathways are also implicated in metastasis” are not original. Please cite the original publication by Harold Dvorak: doi: 10.1056/NEJM198612253152606.

Pg 10, Ln 28. What is meant by “etc…”? Do not assume the reader will know what “etc…” means? Please be specific?

Pg 10, Ln 46. What is meant by “stemness properties”? Do not assume the reader will know what “stemness properties” are? Please elaborate?

Pg 11, Ln 23. The statement “Angiogenesis is mediated by VEGF” is disingenuous. Several other factors are also known to mediate angiogenesis in mammals, namely FGFs, TNFa, TGFb, angiopoietins, PDGF, and some glycosaminoglycans.

Pg 12, Ln 8. The section titled “ROS and stemness” use many concepts and jargon to describe cancer stem cells, “stemness”, hemopoietic stem cells and cell differentiation. This assumes a certain level of compression of the reader, a level that I am not confident all readers possess. Can this section be rewritten using less jargon?

Pg 12, Ln 23. The section titled “4.3. Oxidative stress and metastasis: cellular actors involved” commences with the statement “Macrophages, neutrophils and fibroblasts are major ROS producers in the tumor microenvironment”. However, while the discussion includes macrophages and fibroblasts, neutrophils are not discussed. Please resolve this inconsistency.

Pg 14, Ln 2. What is the word “intertwin”? I presume that this should be ‘intertwine’.

Pg 14, Ln 6. In my understanding, “pathological fibrosis” is associated with absence of cell motility, whereas “cancer cells spread” is associated with enhanced cell motility. I am not convinced that the “slightest disturbance” of macrophage ROS biosynthesis produces opposed outcomes in transformed cells (i.e. cancer) and untransformed cells.

Pg 14, Ln 11. The suggestion that “Controlling oxidative stress level in wound… environment… promote wound healing” is an already well-established wound-management intervention, ‘debridement’.

Pg 14, Figure 3. The vast majority of clinically diagnosed chronic wounds do not result in fibrosis! On the contrary, chronic wounds, also called ‘hard-to-heal’ wounds, fail to re-epithelize (close) due to insufficient (failure) epithelial cell migration and proliferation. Fibrosis is rarely a consequence in cutaneous wounds that fail to close!

Author Response

Responses to Reviewer 2 concerning the manuscript:
Manuscript ID: biomedicines-1953191
Title: Wound healing versus metastasis: role of oxidative stress
Authors: Tatiana Lopez, Maeva Wendremaire, Jimmy Lagarde, Oriane Duquet, Line
Alibert, Brice Paquette, Carmen Garrido, Frédéric Lirussi *
We thank the Reviewer for his/her detailed comments and extensive proofreading of our manuscript. We considered and answered his major and minor comments. We believe that all the requested modifications allowed an easier reading of the manuscript and a better understanding of the reviewed topic.
Pg 2, Ln 8. The statement “Three distinct and overlapping steps are needed in the physiological wound healing process…” is controversial! This is the former perspective, where coagulation (hemostasis) and inflammation are considered as a common phase. The evidence is that coagulation (alt: thrombogenesis) and inflammation are independent events (e.g., ‘sterile inflammation’). Thus, the consensus in the field is that wound healing is a cascade of four, overlapping, phases: hemostasis, inflammation, proliferation, and maturation. See Singer & Clark 1999. doi: 10.1056/NEJM199909023411006; Martin 1997. doi: 10.1126/science.276.5309.75.
Response 1: We changed our proposal to the consensus four overlapping phases. Accordingly, references have been added (Line 48) and modified the sentences (line 49-50).
Pg 2, Ln 11. What is the evidence “Neutrophils and tissue resident macrophages are the first responding cells to the wound”? (citation required)
Response 2: We added references indicating that these immune cells are within the first recruited to the site of injury. (Line 52)
Pg 2, Ln 12. What is the evidence “This stage also induces immune cells invasion particularly monocytes recruited from the bone marrow and differentiated into mature inflammatory macrophages (named M1)”?
Response 3: A reference has been added (Line 54) and the sentence re-phrased (Line 52)
Pg 2, Ln 16. What is the evidence “Resolving anti-inflammatory macrophages (named M2) contribute to remove cells and bacteria debris by efferocytosis or phagocytosis”?
Response 4: A reference has been added (Line 59)
Pg 2, Ln 19. What is the evidence “…tissue remodeling macrophages promote restoration of functional integrity of tissue”?
Response 5: We specified that macrophages induced MMP expression leading to restoration of tissue integrity. We also added a reference. (Line 60,61, 62)
Pg 3. Fig 1. Critical cellular populations are absent from Figure 1. Cutaneous wound healing in humans is primarily orchestrated and mediated by keratinocytes, not just macrophages, lymphocytes and granulocytes! This figure is inaccurate. It is misleading.
Response 6: We agree with referee 2 that keratinocytes play a key role in cutaneous wound healing. Nevertheless, this review was not focused on cutaneous wound healing but wound healing in general. Figure 1 represents the cells involved in all wound healing process, which is why keratinocytes are not included. For illustration, other reviews dealing with wound-healing process in general do not mention keratinocytes (Guillama-Prats The role of MSC in wound healing, scarring and regeneration 2021).
Pg 3. Ln 1. The pathological consequences of “Deficient wound healing due to disturbance at any point in the process” is much wider than merely “pathological fibrosis”. This statement is misleading.
Response 7: As proposed by the reviewer, we changed this statement indicating pathological wounds rather than pathological fibrosis. (Line 64, 65)
Pg 3. Ln 5. Why is the role of macrophages limited to “damage either induced by infection, autoimmune disorders, mechanical or toxic injuries”? As a key ‘actor’ the role of macrophages is ubiquitous to all tissue injuries.
Response 8: We have taking note of this remark. We agree that macrophages are key actors in tissue injuries and we consider that infection, autoimmune disorders, mechanical or toxic injuries represent all the various damage categories.
Pg 4. Ln 1. What is the evidence “…resident macrophages, bone marrow-derived macrophages, along with neutrophils, are among the first cells recruited to the site of injury”?
Response 9: See response 2
Pg 4. Ln 4. What is the “early research” that “highlighted their pro-inflammatory and scavenging contribution…”? (a citation is required).
Response 10: References have been added (Line 77)
Pg 4. Ln 5. What is the evidence “Cellular response is then initiated by secreted inflammatory mediators”?
Response 11: References have been added (Line 79)
Pg 4, Ln 8. What is the “microenvironment stimuli” to which macrophages respond “to remove dead cells and dampen inflammation”? The cited evidence is from hepatocytes (a mesenchymal type). What is the evidence this mechanism is conserved in epithelia, i.e. skin cell type?
Response 12: A reference has been added considering skin wound healing modulation by macrophages (Line 82).
Pg 4. Ln 11. What is the evidence “M2 phenotype, promote… proliferation and blood vessel development through growth factors (PDGF, IGF-1, VEGF) and reduce local hypoxia following injury”?
Response 13: References have been added (Line 86)
Pg 4, Ln 15. What are “stromal cells”? (…presumably they are distinct from fibroblasts.)
Response 14: Stromal cells are multipotent cells that can differentiate into various mesenchymal tissue lineage. They are essential components in lymphoid and non-lymphoid tissues and their role in wound healing have been studied and extensively reviewed for the paste decades.
Pg 4, Ln 16. What are “tissue-remodeling macrophages”? We’ve been introduced to M1 and M2 phenotype macrophages; now we have another macrophage phenotype!
Response 15: The traditional M1/M2 phenotype is a simplified classification to describe in vivo macrophages. The heterogeneity of macrophages has been well described and tissue remodeling macrophages are a supplementary subtype found specifically during the remodeling phase of wound healing and closely related to M2 phenotype. This is now better explained in the revised manuscript (Line 88).
Pg 4, Ln 19. What is the evidence “three functional phenotypes do not involve three distinct macrophage subsets but an activation continuum that evolves, according to cellular ontogenesis and environmental stimuli, from a pro-inflammatory to a resolutive phenotype”? I suggest that the average reader will be confused by this plethora of macrophages phenotypes are but an “activation continuum” responding to “cellular ontogenesis and environmental stimuli”. I am barely keeping up, and I work in this field! As this is a review, please substantiate your interpretations with published evidence.
Response 16: We modified the sentence (Line 91) and references have been added (line 93)
Pg 4, Ln 25. What is the evidence “ROS (superoxide anion [O2•–] and hydrogen peroxide [H2O2]) act in the early phase of wound healing to induce vasoconstriction, platelet activation and defend host from bacterial invasion”?
Response 17: References have been added (Line 100)
Pg 4, Ln 28. What is the evidence “ROS signaling allows the recruitment of neutrophils, macrophages and/or lymphocytes to the site of injury”?
Response 18: This sentence has been modified to indicate ROS chemoattractant properties and a reference has been added (Line 101, 102)
Pg 4, Ln 33. What is the evidence “ROS level is finely controlled by small anti-oxidant molecules…”?
Response 19: A reference has been added (Line 108)
Pg 4, Ln 49. What is the “well documented (evidence) that non-healing wounds, due to diabetes, or chronic wounds characteristic of pathologies such as inflammatory bowel diseases, are associated with a higher ROS level”? (sic)
Response 20: References have been added (Line 124)
Pg 5. Ln 12. What is the “Further evidence…”?
Response 21: References have been added (Line 140)
Pg 5, Ln 13. What is the meaning of “…first main isotypes found…”?
Response 22: We agree that this sentence is confusing. We changed the sentence to “NOX1 and NOX2 are the main isotypes expressed in both bone marrow monocytes and bone marrow-derived macrophages” and added a reference. (Line 141)
Pg 5, Ln 15. having established that there are 3 phenotypic populations of macrophages, which population is referred to in “…human and murine macrophage populations…”?
Response 23: The references cited only indicated human or murine macrophages which refer to immature macrophages. We modified the sentence to emphasize that these are immature macrophages (Line 143)
Pg 5, Ln 18. What is the evidence “ROS, …can modulate macrophages function at various stages”?
Response 24: This sentence introduces the process of ROS modulating macrophages function explained further away with the references.
Pg 5, Ln 6. What is the evidence “ROS are essential for the monocytes to macrophages translation”? What is “monocytes to macrophages translation”?
Response 25: We agree with the reviewer’s remark and modified the sentence to make it clearer for the reader (Line 150). Corresponding references are already in the manuscript (number 51, 52 and 53).
Pg 5, Ln 17. What is the evidence “chemically inhibition of ROS generation may affect the monocyte-macrophage differentiation process”? (sic)
Response 26: The references are already cited (51 and 52)
Pg 5, Ln 22. What is the evidence “ROS produced by neutrophils allow bone marrow monocytes to differentiate into macrophages”?
Response 27: A reference has been added (Line 154)
Pg 5, Ln 26. The statement “ROS inhibition only acts during the differentiation stage and has no effect on the phenotype and function of mature macrophages” contradicts previous statements outlining how ROS inhibitors affect macrophage maturation/differentiation. This also contradicts the subsequent statement (Pg 5, Ln 35) “ROS are required for macrophages differentiation and polarization”.
This reader is confused.
Response 28: To be clearer, we modified the sentence as follows “ROS inhibition only acts during the polarization stage and has no effect on the phenotype and function once the macrophage is mature” (Line 158-159). We also modified the subtitle 2.1. (Line 145)
Pg 5, Ln 36. The section “3.2. NOXs in macrophages polarization” is predicated on the assumption that murine and human monocyte /macrophage populations possess equivalent genotypes, phenotypes, functional types, and respond equivalently to ReDox-dysregulated tissue environments.
What is the evidence that supports this assumption in human skin tissue?
Response 29:
As indicated above, our review was not specifically focused on human skin but on the common hallmarks of wound healing and similarities with metastatisation. We assume that some general assertions may be a little less relevant in some specific microenvironments.
Pg 6, Ln 25. What is the evidence “hypoxia, inflammation and oxidative stress… are regulated by numerous transcription factors”? This is oversimplification and ignores fundamentals of chemistry.
Response 30: We agree that this sentence oversimplifies the regulation of hypoxia, inflammation and oxidative stress but the purpose of this review was not to decipher the signaling pathways involved in the regulation of these three processes. This was just an introductory sentence to discuss the three main transcription factors in macrophages essential in the context of oxidative stress in wound healing. We have now added “among other things” to moderate our assertion (Line 204).
Pg 7, Figure 2. “ROS signaling pathways involved in wound healing and metastasis.”
What is the tubular structure adjacent to (1) (top left)?
There is some confusion in Ln 5 of the legend. It is stated that Nrf2 binding to ARE binding “inhibits… anti-oxidant defense expression” yet is illustrated as enabling transcription of NQO-1, HO-1, SOD, CAT, GPx, NOX4?
Figure 2 includes many abbreviations that are not defined herein, nor in the main text.
Response 31: We agree with this remark and we acknowledge for this misunderstanding figure. We added the name of the tubular structure, which is the proteasome complex. We also modified the legend, which was confusing, to indicate that Nrf2 enhances the anti-oxidant defense response (line 1044). We also have been attentive to the abbreviations and added a table of acronyms on editor’s request.
Pg 8, Ln 8. What is the evidence “In wound, local oxygen level is reduced due to blood vessel destruction”? (sic)
It is my understanding the opposite is true! Skin tissue has an O2 pp of approx. 8.0 ±3.2 mm Hg. The O2 of air (at sea level) is 160 mm Hg. Surface oxygen is ~20 times greater, and triggers many events associated with tissue injury! I think the authors are referring to the sub-cutaneous wound bed.
Response 32: The reviewer’s remark is very interesting. Indeed, we are only presenting the general wound healing process without specifying the damaged tissue. We agree that in skin tissue, local hypoxia may be less important. To justify our statements, a reference has been added (Line 211).
Pg 8, Ln 9. What is the evidence “…local hypoxia implicates macrophages and induces ROS production”?
Response 33: References have been added (Line 214)
Pg 8, Ln 37. What is “a positive regulation of HIF-1”?
Response 34: This sentence has been modified and we indicated that Li et al demonstrated the critical role of HIF-1α during macrophages differentiation (Line 239-240).
Pg 8, Ln 52. What is the evidence “NF-kB is a homo- and hetero-dimeric complex resulting from the five monomers”?
Response 35: NF-kB structure has been extensively studied but we agree that a reference is missing. Therefore, we added a reference (Line 257) and we specified that these monomers have only been found in mammals.
Pg 9, Ln 20. What is the evidence “Nrf2 is a basic leucine zipper (bZIP), which is a major sensor for oxidative stress”?
Response 36: A reference has been added (Line 279)
Pg 10, Ln 16. The references cited in support of the statement “Many wound healing cellular actors, molecular mechanisms and signaling pathways are also implicated in metastasis” are not original. Please cite the original publication by Harold Dvorak: doi: 10.1056/NEJM198612253152606.
Response 37: The reference has been added (line 329).
Pg 10, Ln 28. What is meant by “etc…”? Do not assume the reader will know what “etc…” means? Please be specific?
Response 38: We completed this sentence by adding “extracellular matrix remodeling, epithelial–mesenchymal transition, metastasis, cancer stem cell maintenance and immune evasion.” (line 340 to 342).
Pg 10, Ln 46. What is meant by “stemness properties”? Do not assume the reader will know what “stemness properties” are? Please elaborate?
Response 39: We acknowledge for this shortcut and explained these properties in the manuscript for the average reader. We added this sentence (line 360 to 362): “Stemness is the ability of a cell to perform self-renewal and is capable of pluripotency. This is an important feature for supplying material for wound closure and for the establishment of cancer cells at the metastatic sites.
Pg 11, Ln 23. The statement “Angiogenesis is mediated by VEGF” is disingenuous. Several other factors are also known to mediate angiogenesis in mammals, namely FGFs, TNFa, TGFb, angiopoietins, PDGF, and some glycosaminoglycans.
Response 40: We tempered this assumption and modified the sentence as “angiogenesis is mainly mediated by VEGF” (line 390).
Pg 12, Ln 8. The section titled “ROS and stemness” use many concepts and jargon to describe cancer stem cells, “stemness”, hemopoietic stem cells and cell differentiation. This assumes a certain level of compression of the reader, a level that I am not confident all readers possess. Can this section be rewritten using less jargon?
Response 41: Many reviews on metastasis use these concepts. We prefer not to modify this paragraph so as not to overload the manuscript. Nevertheless, we clarified the stemness properties (see response 39).
Pg 12, Ln 23. The section titled “4.3. Oxidative stress and metastasis: cellular actors involved” commences with the statement “Macrophages, neutrophils and fibroblasts are major ROS producers in the tumor microenvironment”. However, while the discussion includes macrophages and fibroblasts, neutrophils are not discussed. Please resolve this inconsistency.
Response 42: As mentioned in lines 443-444, we focused on macrophages and fibroblasts since neutrophils activation in wound healing and metastasis has been already extensively reviewed elsewhere.
Pg 14, Ln 2. What is the word “intertwin”? I presume that this should be ‘intertwine’.
Response 43: This taping mistake has been corrected (line 488).
Pg 14, Ln 6. In my understanding, “pathological fibrosis” is associated with absence of cell motility, whereas “cancer cells spread” is associated with enhanced cell motility. I am not
convinced that the “slightest disturbance” of macrophage ROS biosynthesis produces opposed outcomes in transformed cells (i.e. cancer) and untransformed cells.
Response 44: ROS may not have an action on motility per se but on the other hallmarks of these 2 pathologies, for example the action of ROS on remodelling mediated by MMPs. This is better stated in the revised manuscript (Line 414)
Pg 14, Ln 11. The suggestion that “Controlling oxidative stress level in wound… environment… promote wound healing” is an already well-established wound-management intervention, ‘debridement’.
Response 45: We meant that controlling oxidative stress level could be an interesting treatment before resorting to the mechanical act of debridement. Moreover, controlling oxidative stress level may be a complementary strategy (please, see Line 499-500 revised manuscript).
Pg 14, Figure 3. The vast majority of clinically diagnosed chronic wounds do not result in fibrosis! On the contrary, chronic wounds, also called ‘hard-to-heal’ wounds, fail to re-epithelize (close) due to insufficient (failure) epithelial cell migration and proliferation. Fibrosis is rarely a consequence in cutaneous wounds that fail to close! Response 46: We agree with the reviewer. We modified the figure 3 by replacing “fibrosis” by “non-healing wound” (Line 1065)

Reviewer 3 Report

Dear Authors,

I have read the manuscript and I send you my comments:

1) please add the methods used to select the reference

2) Pag 11 you describe the role of MMPs, but some references are missing please add Serra et al., Int Wound J. 2017 Feb;14(1):233-240;

de Franciscis et aL., Int Wound J. 2016 Dec;13(6):1237-1245. d

3) The paper is very well written, the text is clear and the topic may be of interest. The conclusions are consistent with the evidence 

Author Response

Responses to Reviewer 3 concerning the manuscript:
Manuscript ID: biomedicines-1953191
Title: Wound healing versus metastasis: role of oxidative stress
Authors: Tatiana Lopez, Maeva Wendremaire, Jimmy Lagarde, Oriane Duquet, Line
Alibert, Brice Paquette, Carmen Garrido, Frédéric Lirussi *
We thank the Reviewer for his/her positive comments and careful proofreading of our manuscript.
I have read the manuscript and I send you my comments:
1) please add the methods used to select the reference
Response 1: We added this sentence (lines 1013) “This literature review was based on searches on PubMed, web of science, Springer, and Wiley databases, with no time limit but giving preference to recent articles.”
2) Page 11 you describe the role of MMPs, but some references are missing please add Serra et al., Int Wound J. 2017 Feb;14(1):233-240; de Franciscis et aL., Int Wound J. 2016 Dec;13(6):1237-1245.
Response 2: We added these two references at line 62.
3) The paper is very well written, the text is clear and the topic may be of interest. The conclusions are consistent with the evidence
